# Development and Validation of Two-Step Prediction Models for Postoperative Bedridden Status in Geriatric Intertrochanteric Hip Fractures

**DOI:** 10.3390/diagnostics14080804

**Published:** 2024-04-11

**Authors:** Kantapon Dissaneewate, Pornpanit Dissaneewate, Wich Orapiriyakul, Apipop Kritsaneephaiboon, Chulin Chewakidakarn

**Affiliations:** 1Department of Orthopedics, Faculty of Medicine, Prince of Songkhla University, Hat Yai 90110, Thailand; dpornpanit@yahoo.com (P.D.); poknitting@hotmail.com (W.O.); kapipope@yahoo.com (A.K.); chulinccc@yahoo.com (C.C.); 2Department of Clinical Research and Medical Data Science, Faculty of Medicine, Prince of Songkla University, Hat Yai 90110, Thailand

**Keywords:** hip fracture, bedridden, prediction model, rehabilitation

## Abstract

Patients with intertrochanteric hip fractures are at an elevated risk of becoming bedridden compared with those with intraarticular hip fractures. Accurate risk assessments can help clinicians select postoperative rehabilitation strategies to mitigate the risk of bedridden status. This study aimed to develop a two-step prediction model to predict bedridden status at 3 months postoperatively: one model (first step) for prediction at the time of admission to help dictate postoperative rehabilitation plans; and another (second step) for prediction at the time before discharge to determine appropriate discharge destinations and home rehabilitation programs. Three-hundred and eighty-four patients were retrospectively reviewed and divided into a development group (*n* = 291) and external validation group (*n* = 93). We developed a two-step prediction model to predict the three-month bedridden status of patients with intertrochanteric fractures from the development group. The first (preoperative) model incorporated four simple predictors: age, dementia, American Society of Anesthesiologists physical status classification (ASA), and pre-fracture ambulatory status. The second (predischarge) model used an additional predictor, ambulation status before discharge. Model performances were evaluated using the external validation group. The preoperative model performances were area under ROC curve (AUC) = 0.72 (95%CI 0.61–0.83) and calibration slope = 1.22 (0.40–2.23). The predischarge model performances were AUC = 0.83 (0.74–0.92) and calibration slope = 0.89 (0.51–1.35). A decision curve analysis (DCA) showed a positive net benefit across a threshold probability between 10% and 35%, with a higher positive net benefit for the predischarge model. Our prediction models demonstrated good discrimination, calibration, and net benefit gains. Using readily available predictors for prognostic prediction can assist clinicians in planning individualized postoperative rehabilitation programs, home-based rehabilitation programs, and determining appropriate discharge destinations, especially in environments with limited resources.

## 1. Introduction

Geriatric hip fractures remain a major health concern globally, with estimations projecting the emergence of nearly 6.3 million cases annually by 2050, 46% of which are intertrochanteric fractures [1,2]. Demographic characteristics and outcomes differ between patients with femoral neck and intertrochanteric fractures, with the latter group being older and frailer, leading to higher rates of poor outcomes [2,3,4]. Approximately 12–25% of patients with intertrochanteric fractures become immobilized and bedridden after three months [2,5]. To prevent bedridden status in patients with hip fractures, various rehabilitation programs have been developed, including early ambulation training, high-intensity physiotherapy with multiple daily sessions, and home rehabilitation programs [6,7,8,9]. These interventions effectively help to prevent patients from becoming bedridden after hip fracture treatment [8]. However, implementing such programs for all patients may be financially unfeasible on a public healthcare level because of the associated costs and human resource demands. Classifying high-risk patients can help optimize resource allocation, reduce the cost of implementing rehabilitation programs, and alleviate the workload on healthcare workers by minimizing unnecessary interventions for low-risk patients.

Only a few prediction models have been specifically developed for predicting ambulatory status in intertrochanteric fractures [10,11]. Tomita et al. developed a model to predict the bedridden status after three months using patients’ postoperative status at two weeks [10]. Meanwhile, models from Adulkasem et al. predicted the ambulatory status based on the quality of fracture reduction during the operation, allowing surgeons to improve reduction in real time based on predictions [11]. These previously developed models also have limitations in terms of prediction timing and barriers to adoption, such as model complexity and methods for obtaining intraoperative measurements.

Studies have identified several readily accessible predictors of poor ambulatory status following hip fracture, including age, dementia, American Society of Anesthesiologists (ASA) classification, pre-fracture ambulatory status, and ambulatory status before discharge [12,13,14,15,16,17]. These factors were consistently correlated with ambulatory outcomes during follow-up. Combining these predictors into a simple prediction model could possibly help categorize patients at high risk of becoming bedridden.

Our goal was to create a two-step prediction model using simple, easily retrievable predictors reported in the literature to assist clinicians in deciding on care during hospital stay and after discharge. We hypothesized that simple prediction models, integrating these readily available predictors, could accurately categorize patients at high risk for becoming bedridden after surgery. The two-step model comprises two individual models for predicting 3-month postoperative bedridden status at different timepoints during the management of intertrochanteric hip fractures: the first predicts during the preoperation period to help clinicians determine if the patient requires a specialized postoperative rehabilitation program; and the second predicts at discharge to assist in determining the discharge destination and whether the patient needs a specialized home rehabilitation program to reduce the risk of becoming bedridden.

## 2. Materials and Methods

### 2.1. Data Source and Participants

This study is reported according to the Transparent Reporting of a multivariable prediction model for individual prognosis or diagnosis (TRIPOD) statement (Appendix A). We retrospectively reviewed the electronic medical records (EMRs) of patients with intertrochanteric fractures treated with cephalomedullary nails at a single university hospital in southern Thailand over a 12-year period, with all patients having a minimum follow-up of 3 months. Data were collected from two patient cohorts. The development cohort comprised patients treated between January 2010 and December 2020, and the validation cohort comprised patients treated between January 2021 and December 2022.

Patient selection was based on the following criteria:

Inclusion criteria:-Aged at least 60 years;-Sustained an isolated intertrochanteric hip fracture resulting from low-energy trauma;-Underwent treatment with a cephalomedullary nail.

Exclusion criteria:-Lost to follow-up before the 3-month follow-up;-Bedridden prior to the hip fracture occurrence;-Mortality before the 3-month follow-up.

The treatment protocol for fragility intertrochanteric fractures was as follows. The preoperative evaluation was initially conducted in the orthopedic trauma unit. Internist consultations were arranged for patients with specific medical conditions, such as cardiologist consultations for patients with a history of ischemic heart disease or neurologist consultations for those with previous ischemic stroke. After evaluation by all relevant units, the patients were scheduled to undergo surgery at the earliest available slot.

All surgeries were performed by the attending orthopedic trauma surgeons or under their supervision. A daily 30-min physical therapy session was provided postoperatively. The physical therapy regimen included triceps and quadriceps exercises, deep breathing exercises using an incentive spirometer, and ambulation training. Patients who could not walk using a walker were trained in wheelchair transfer with at least one caregiver. Follow-up appointments were scheduled at two weeks, six weeks, and three months postoperatively.

### 2.2. Outcome and Predictors

The primary outcome was the bedridden status after surgery assessed three months postoperatively. We routinely recorded the patients’ ambulatory status at our trauma follow-up clinics; patients were considered bedridden if it was recorded in the EMR.

We predefined four preoperative predictors: age, diagnosis of dementia, pre-fracture ambulatory status, and ASA classification. One predischarge predictor was considered: ambulation status before discharge. Age was defined as the patient’s age in years at the date of injury as recorded in the EMR. Dementia was defined as a known dementia status recorded in the EMR before or at the time of admission by one of the attending physicians. The pre-fracture ambulatory status was divided into two categories: completely independent patients who could independently ambulate indoors, outdoors, and in the community without any use of a gait aid prior to fracture, while gait aid/indoor ambulation included patients who were independently ambulating in the community or outdoors with some use of a gait aid or patients who only independently ambulated at home but were unable to travel outside independently. The attending anesthesiologist determined the ASA classification before surgery for each case. Ambulatory status before discharge was defined as that recorded in the discharge summary on the date of discharge. We categorized the ambulatory status before discharge into three groups: walking with a walker, being able to sit independently but unable to walk using a walker, and being unable to sit independently. The baseline reference was a 60-year-old patient without dementia who was totally independent prior to the fracture, was determined to have ASA class II, and could walk with a walker before discharge.

### 2.3. Statistical Analysis

We defined five predictors with six degrees of freedom (DFs), thus requiring at least 60 events of interest for prediction model development, with the remaining data used for external validation. Statistical analyses were performed based on a complete-case analysis, and patients lost to follow-up were excluded, as previously stated. Comorbidities not recorded in the EMR were assumed to be negative.

All statistical analyses were performed using R programming language version 4.1.3 (R Foundation for Statistical Computing, Vienna, Austria). Comparisons between the development and validation cohorts were performed using a two-sample *t*-test for normally distributed continuous variables, the Wilcoxon rank sum test for non-normally distributed continuous variables, and a chi-square test for categorical variables. Univariate analysis of the association between predictors and outcomes was not performed. Statistical significance was set at *p* < 0.05.

We developed two multivariable logistic regression models to predict the bedridden status. The preoperative model had four predictors: age in years (continuous, 1 DF), dementia (categorical, 1 DF), pre-fracture ambulatory status (categorical, 1 DF), and ASA classification (1 DF). The predischarge model had the same predictors as the preoperative model, with an additional predictor: ambulatory status before discharge (categorical, 2 DFs). Internal validation of the development dataset was performed using 1000 bootstrap resamples. The bootstrap performance was calculated using the 0.632 method. To determine the temporal relevance of the model, external validation with 1000 bootstrap resamples was performed on the validation dataset containing the most recently treated cohort at our institution.

The model performance was evaluated using three main metrics. Model discrimination, which assesses the ability to distinguish between patients who will become bedridden and those who will not, was determined using the area under the receiver operating characteristic curve (AUC). Successful prediction model development was defined as a model with an AUC score of 0.7 or more. Model calibration, which measures how accurately the estimated risk of becoming bedridden aligns with the observed frequency of bedridden cases, was evaluated using calibration plots and the Hosmer–Lemeshow goodness-of-fit test. Finally, clinical usefulness was assessed through a decision curve analysis within a cut-off range of 10–35%.

The decision curve analysis (DCA) was first described by Andrew Vickers and Elena Elkin in 2006 [18]. The fundamental concept of DCA is to assess the net benefit of various decision-making strategies across a spectrum of threshold probabilities. A threshold probability is defined as the point at which a patient or doctor feels indifferent about initiating treatment, with the underlying assumption being that these probabilities vary among patients and doctors, reflecting individual preferences for balancing the benefits of true positives against the harms of false positives. For a prediction model to be practical, its expected benefit should surpass its expected harm at each sensible threshold probability.

Two types of decision curves were analyzed: the net benefit curve, comparing the net benefit at different potential risk thresholds for intervention using the prediction model with the policies of “treat all patients” or “treat none of the patients”; and the net reduction in intervention, which expresses the net benefit as true negatives to compare the rate of unnecessary interventions avoided by utilizing the prediction model in relation to the “treat all patients” policy.

The net benefit is the difference between the expected benefit and harm. For any given model or strategy, the net benefit is calculated as follows:-Expected Benefit (EB) is the number of true positives (TP) divided by the total number of patients (N);-Expected Harm (EH) is the number of false positives (FP) divided by N, which is then adjusted for the threshold probability’s odds (p/(1 − p)).

The equation for net benefit (NB) at each threshold probability (p) is:Net Benefit = (TP/N) − (FP/N) × (p/(1 − p))

The net reduction in intervention quantifies how the model reduces the rate of unnecessary interventions relative to the “treat all” approach. For any given model or strategy, the net reduction in intervention is calculated as follows:Net Reduction in Intervention = (Net Benefit of the model − Net Benefit of the “treat all patients” policy) × ((1 − p)/p)

## 3. Results

In total, 437 patients were assessed for eligibility. Among these patients, 34 were lost to follow-up, 13 were bedridden before the fractures, and 6 died before the three-month follow-up, leaving 384 patients for analysis. Patients were divided into two cohorts. The development cohort comprised 291 patients treated at our institution between January 2010 and December 2020. The validation cohort comprised 93 patients treated between January 2021 and December 2022 (Figure 1).

### 3.1. Patient Characteristics

The overall median patient age was 83 (78–87) years (Table 1). Most participants (70%) were women. One hundred and seventy-nine patients (47%) were completely independent before the fracture, and 205 (53%) were independent but required the use of a gait aid or could only ambulate indoors. Two hundred forty-three patients (63%) were classified as ASA class III. Only 29 (7.6%) patients had been previously diagnosed with dementia. Most patients (62%) could walk using a walker postoperatively before discharge, 32% could sit independently but could not walk with a walker, and 6% could not sit independently before discharge. Overall, 86 of the 384 patients (22%) became bedridden three months postoperatively.

The development and validation cohorts were comparable in terms of sex, body mass index (BMI), ASA classification, comorbidities, ambulatory status before discharge, and outcomes. However, patients in the validation cohort had a slightly higher age (85 years (79–89) vs. 83 years (77–86), *p* = 0.005) and a more significant proportion of patients who used gait aids or had indoor ambulation (70% vs. 48%, *p* < 0.001). In the validation cohort, 43% of the patients had a simple A1 fracture according to the AO/OTA classification compared with 26% in the development cohort (*p* = 0.011). Most patients (87%) in the validation cohort underwent surgery within 72 h of admission. In contrast, only 70% of the development cohort underwent surgery within 72 h. The overall median length of stay after operation was 5 (4–8) days.

### 3.2. Model Development and Model Specification

Two multivariate logistic regression models were derived from the development cohort to predict the probability of a bedridden status at three months postoperatively. Because all predictors were predefined, predictor selection techniques were not employed during the analysis.

The specifications of the derived models are listed in Table 2. The variance inflation factors (VIFs) for all factors in both models were less than 1.5, indicating no significant multicollinearity among the predictor variables. In the preoperative model, the regression coefficients for all factors indicated a significant association with bedridden status at three months postoperatively, with all *p*-values < 0.05.

All predictors had a positive correlation with bedridden status after including ambulatory status before discharge in the predischarge model; however, the correlation of ASA classification and dementia was not statistically significant (*p* = 0.3 and 0.2, respectively). Ambulatory status before discharge exhibited the strongest association with the bedridden status. The coefficient for sitting independently at discharge was 2.12 (95% CI 1.39–2.90) compared with being able to walk using a walker before discharge. The coefficient of the inability to sit independently before discharge was 2.95 (95% CI 1.79–4.21).

The equation for the preoperative model’s linear predictor is
LP_preop_ = −7.39 + (0.06 × Age) + (0.72 × ASA III) + (0.94 × Dementia) + (1.14 × Pre-fracture Gait Aid/Indoor Ambulation)

The equation for the predischarge model’s linear predictor is
LP_predischarge_ = −8.65 + (0.06 × Age) + (0.40 × ASA III) + (0.62 × Dementia) + (0.81 × Pre-fracture Gait Aid/Indoor Ambulation) + (2.12 × Sit independently before discharge) + (2.95 × Cannot sit independently before discharge)

To convert the linear predictor into the probability of bedridden status at 3 months, apply the inverse logit function to the linear predictor:P(Bedridden) = 1/(1 + exp(−LP))

### 3.3. Model Performance: Area under the ROC Curve

In Table 3, the discriminative capacities of the two prediction models are quantified by the area under the receiver operating characteristic curve (AUC). Apparent performance demonstrates the AUC of the model when evaluated on the development set. Validation performances were derived from 1000 bootstrap samples; internal validation was performed on the development set, and external validation was performed on the validation set.

The preoperative model demonstrated an apparent AUC of 0.76 (CI: 0.68–0.83), with internal and external validation AUCs of 0.74 (CI: 0.68–0.80) and 0.72 (CI: 0.61–0.83), respectively. The predischarge model showed superior discrimination with an apparent AUC of 0.85 (CI: 0.79–0.91), with the internal and external validations yielding AUCs of 0.84 (CI: 0.79–0.89) and 0.83 (CI: 0.74–0.92), respectively.

### 3.4. Model Calibration

The calibration plots are shown in Figure 2. Calibration of the preoperative model showed slight overall miscalibration in both the development and validation sets. During external validation, the preoperative model provided slightly higher predicted probabilities than observed probabilities. Conversely, the predischarge model displayed near-perfect calibration on the development set, with minimal miscalibration; each point of the grouped predictions aligned closely with the dotted line, representing equal predicted and observed probabilities. External validation of the predischarge model indicated a slight overestimation in the high predicted probability group; however, it remained well calibrated for groups with predicted probabilities of less than 50%.

Comparing the preoperative and predischarge models, the latter displayed a broader and more spread-out distribution of probability estimates. The Hosmer–Lemeshow test for goodness-of-fit did not identify significant miscalibration in any calibration plot.

### 3.5. Decision Curve Analysis

The net benefit and net reduction in interventions were plotted across a theoretical cutoff threshold range of 10–35% (Figure 3). The curves in Figure 3 show the highest net benefit and net reduction in interventions for the predischarge model, followed by the preoperative model. This indicates the usefulness of both models over the default policies of “treat all” or “treat none”. The predischarge model is shown to be superior to the preoperative model in net benefit gains and net reduction in intervention across all considered threshold probabilities.

Interpreting decision curves involves understanding how net benefit and net reduction in intervention curves relate to clinical decisions. The standard policies might be “treat all”—providing every patient with specialized postoperative intervention to prevent bedridden status—or “treat none”, where all patients receive only the standard postoperative protocol. Since different clinicians and patients have varying values, their threshold probability for treatment differs. For prediction models to be truly useful, they need to offer benefits across all reasonable threshold probabilities. This means that models demonstrating higher net benefit or net reduction in interventions at all threshold probabilities could theoretically be useful for all potential users of the model. This is illustrated by our models in Figure 3, which show higher net benefit and net reduction in interventions across the considered thresholds.

The specific cut-off threshold range of 10–35% was selected based on a reasonable “harm-to-benefit” ratio in relation to the policy of providing additional treatment to patients at risk of becoming bedridden at three months. A 10% cutoff signifies a willingness to administer additional interventions to ten patients to prevent one patient from becoming bedridden at three months (equating to a ratio of one true positive to nine false positives). A cut-off threshold of 35% roughly reflects the willingness to administer additional interventions to three patients to prevent one patient from becoming bedridden at three months (resulting in a ratio of 1:2). Note that in this context, “harm” refers to the financial and personal burden placed on the hospital when implementing interventions for the patients rather than harm to the patients themselves.

To clarify, the “net benefit” reflects the proportion of true positives gained from the intervention minus the proportion of false positives based on a specific threshold probability that a patient will become bedridden. It is calculated by comparing the proportion of correctly identified patients who benefit from the intervention with those incorrectly identified when no intervention is applied. In practical terms, a net benefit of 13% at a cut-off value of 20% for the predischarge model indicates that for every 100 patients, 13 will correctly receive the necessary intervention to prevent them from becoming bedridden, after accounting for the trade-off of potentially intervening for patients who would not benefit.

Similarly, the “net reduction in intervention” denotes the reduction in the number of unnecessary interventions when using the predictive model as opposed to a universal treatment policy. This metric is defined by the percentage of patients who avoid unnecessary interventions without overlooking those who would benefit. In our validation cohort, a net reduction in intervention of 35% means that with the predischarge model applied at the 20% cut-off, we can avoid unnecessary interventions for 35 out of every 100 patients compared with the default strategy of treating all patients.

## 4. Discussion

This study successfully developed well-calibrated prediction models that demonstrated exemplary performance in predicting bedridden status three months after the operative treatment of intertrochanteric fractures. The preoperative model included four predictors: age, dementia, ASA classification, and pre-fracture ambulatory status. To create a predischarge model, ambulatory status before discharge was incorporated into the preoperative model. After adding ambulatory status before discharge to the preoperative model, we found a significant enhancement in the model’s ability to predict the bedridden status. This improvement is evident in terms of discrimination, calibration, and overall usefulness. This study also reaffirms that these five predictors are associated with postoperative bedridden status after hip fracture surgery [12,14,15,17,19].

The models performed well, with only a minimal decrease in the AUC observed after internal and external validations. These findings suggest that the models were not overfitted and could potentially be generalized to similar populations [20]. The calibration plots further confirmed the reliability of the models, as they demonstrated good calibration not only in the internal validation but also in the external validation process. Considering that different physicians may have different treatment thresholds influenced by factors such as economic considerations, public policies, or hospital resources, good calibration with reliable risk estimation for each patient will play a vital role in informed, shared decision-making and evidence-based policymaking. However, this aspect has often been overlooked in developing clinical prediction models [21].

We further demonstrate the usefulness of the models using decision curves. Existing evidence has highlighted the benefits of intensive rehabilitation programs (2–3 sessions per day) and home-based rehabilitation programs, which significantly improve mobility in patients with hip fractures [7,8,22]. Although these interventions should ideally be extended to all patients, practical constraints on available resources often limit their implementation. At our institution, the postoperative rehabilitation protocol involves a single daily 30-min session of physical therapy. Establishing a postoperative rehabilitation protocol aimed at reducing the prevalence of bedridden status and adopting an intensive rehabilitation approach for all patients would result in a threefold increase in the workload of physical therapists dedicated to patients with hip fractures. Consequently, this will reduce the time available to attend to other patient categories. Utilizing the preoperative model to identify patients suitable for intensive rehabilitation programs within the 10–35% threshold yields positive net benefits, and net intervention can be avoided, surpassing the approach of implementing the new protocol for all patients. Similarly, applying the predischarge model to select patients for home-based rehabilitation programs also resulted in positive outcomes. The chosen range for treatment thresholds, ranging from 10 to 35%, was determined by considering patients with odds of becoming bedridden less than 1:9 (risk < 10%) as too resource-intensive to treat and those with odds of 1:2 (risk > 35%) as those for which treatment is necessary. The selection of thresholds within the 10–35% range can be tailored based on the resource availability specific to each hospital, involving a trade-off to ensure optimal utilization.

While previous studies have developed various models to predict ambulatory status post-hip fracture surgery, our study contributes to this body of work by providing a novel perspective through the use of simple, clinically obtainable predictors at two critical junctures in patient care: admission and predischarge. This approach can inform treatment protocols during hospitalization and shape post-discharge rehabilitation plans, including decisions about the most suitable discharge destination.

Compared with the model developed by Kim et al., which provides a prediction for ambulatory status at one-month post-surgery, our model extends the prediction to three months [23]. While prediction at one month may be beneficial for short-term care planning, it might not fully account for the variations in recovery trajectories that can manifest beyond the acute postoperative phase. Our three-month prediction window aligns with evidence suggesting that substantial functional gains can be achieved with rehabilitation interventions administered beyond the first month, with the majority of physical activities of daily living (PADL), including mobilizing ability, recovering and plateauing by 3 months [24].

In contrast to the decision tree model developed by Tomita et al., which predicts independent ambulatory status at three months using the Barthel index (BI) at two weeks postoperatively and dementia status as predictors [10], our models provide early predictions at the time of admission. The need to wait for two weeks to assess BI, as required by their model, can delay intervention decisions and may miss the opportunity for immediate postoperative care planning. Our approach mitigates this limitation by incorporating the predictors available at admission, enabling timely evaluations and allowing for more prompt and potentially more effective interventions.

Adulkasem et al. proposed a two-step prediction model focusing on one-year ambulatory status, which included preoperative factors such as age, pre-injury new mobility score, Charlson comorbidity index, and BMI, followed by an intraoperative model integrating eight additional surgical measurements [11]. While their method achieved good predictive accuracy, our models address a different need by providing earlier risk assessments for postoperative bedridden status at three months using easily obtainable clinical predictors at the time of admission and before discharge. This not only offers a valuable timeframe for clinical intervention but also circumvents the variability introduced by relying on intraoperative measurements, which can differ significantly with the type of C-arm machine used and the surgical setting [25].

Despite the significant advantages of our study compared with previous models, it is important to acknowledge several limitations. First, the retrospective nature of the study introduces the potential for bias. For the ambulatory statuses recorded in the electronic medical records, we did not use a standardized ambulatory assessment questionnaire. Instead, ambulatory assessment was recorded as either totally independent or requiring indoor ambulation/use of a gait aid only, which may not fully capture the entire spectrum of ambulatory statuses in the population. Furthermore, despite adhering to standardized recording protocols for hip fracture patients, certain factors, such as the diagnosis of dementia, may not have been consistently recorded. This could lead to the underreporting of cases with dementia, potentially affecting the effectiveness of the predictors. Secondly, our study excluded many patients who were lost to follow-up. Furthermore, competing risks such as mortality before the three-month mark could introduce bias into the model coefficients as a whole. Nevertheless, the temporal external validation demonstrated promising outcomes in our specific population. Finally, even though the models were developed using established predictors, it is important to note that they were ultimately derived from a relatively small dataset obtained from a single university hospital. The risk of becoming bedridden varies among populations. For optimal use, the models should be externally validated for the intended population, and the coefficients should be recalibrated before implementation.

Even though this study successfully achieved its goal of predicting bedridden status at 3 months postoperatively, there remain many opportunities for improvement. Firstly, the benefits of risk stratification and interventions for high-risk patients need to be evaluated after the implementation of the prediction model. While it is assumed that patients at a higher risk of becoming bedridden could benefit more from specialized interventions, some interventions may be futile for patients at a very high risk due to their limited functional reserve, which could not be altered even with intervention. Secondly, clinical prediction modeling is an iterative process. After implementing the model into clinical practice and incorporating multiple interventions to ensure that high-risk patients are treated accordingly, the model could be updated to include a range of differing interventions based on the various costs and resources needed. This could further support shared decision making between clinicians and patients. Lastly, this study focused on a small number of easily obtainable predictors available in standard practice. One area of improvement could be to incorporate more relevant predictors previously studied, such as an evaluation of nutritional status or sarcopenia [26,27,28]. Future research could explore the advantages of incorporating these predictors to enhance the model’s discriminative capacity.

To ensure easy adoption and implementation, we have included graphical score charts for each model illustrating the predicted risk for patients aged between 60 and 100 years. To illustrate the practical utility of these models, we considered a scenario involving a 75-year-old female with an intertrochanteric fracture. This patient was previously an independent indoor ambulator and was not diagnosed with dementia, with ASA class III status based on an evaluation by the attending anesthesiologist during the preoperative assessment. According to the model predictions (Figure 4), this patient had a 22% calculated risk of becoming bedridden at three months. After consulting with the patient and their family members, the attending surgeon enrolled the patient in the newly introduced intensive postoperative rehabilitation program. With efforts from the multidisciplinary team, the patient could sit independently but could not walk with a walker by postoperative day 7. This date was marked as the scheduled discharge date, primarily driven by the lack of inpatient beds available for other trauma patients. Extending the hospital stay for continued physical therapy until the patient could walk using a walker was not a viable option at our institution. The attending surgeon discussed the updated predicted risk of 38% from the predischarge model (Figure 5) with the patient and family. Two options were presented to improve the patient’s chances of regaining independent ambulation. The patient could be transferred to a primary care hospital with a specialized rehabilitation facility or opt for personalized home-based physical therapy sessions provided by a team of physical therapists and nurses.

## 5. Conclusions

In conclusion, we have developed and temporally validated two prediction models to predict the risk of postoperative bedridden status at three months in patients with intertrochanteric fractures. Both models demonstrated acceptable discrimination, were well calibrated, and proved useful across the entire range of cutoff values. We confirmed that simple prediction models using age, dementia status, pre-fracture ambulatory status, ASA, and ambulation status before discharge can accurately categorize patients at high risk of becoming bedridden three months post-surgery. Selecting high-risk patients according to the model for additional interventions, during admission and after discharge, could potentially enhance their quality of life while avoiding the overutilization of healthcare resources, especially in regions with limited capacity.

## Figures and Tables

**Figure 1 diagnostics-14-00804-f001:**
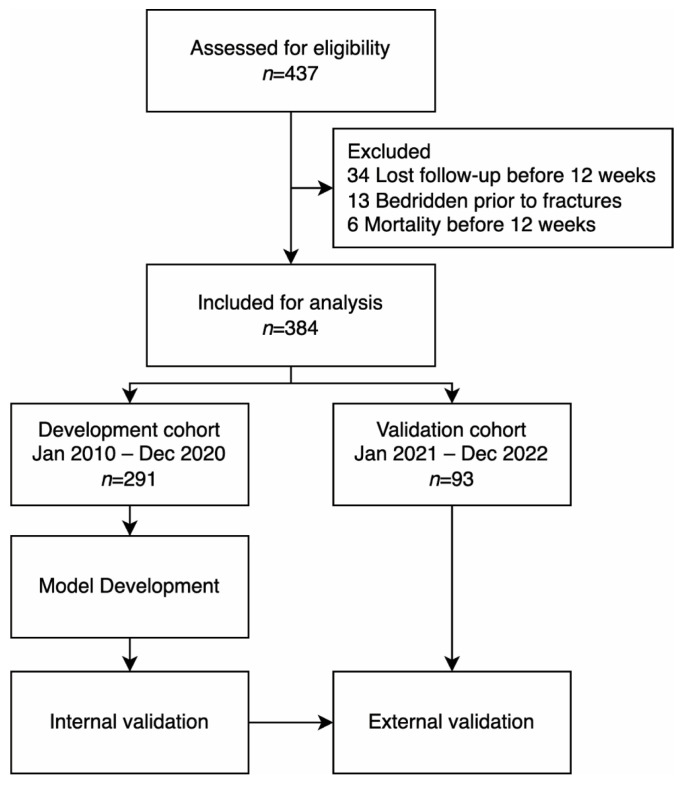
Study flow diagram.

**Figure 2 diagnostics-14-00804-f002:**
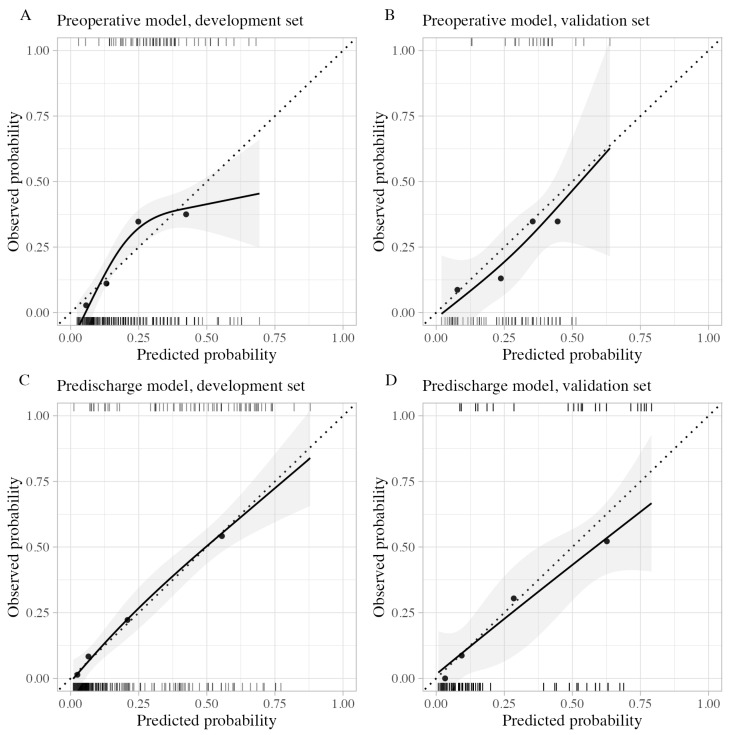
Calibration plots of actual outcome versus predictions on the development and validation cohorts, presented with four quantile groups and smoothed curves using restricted cubic splines with three knots. Solid lines show the calibration curve, dashed lines represent perfect calibration, and circles denote the observed event proportion in quantiles. The grey area reflects the 95% confidence interval for predicted probabilities. Rugs at the bottom indicate participants without bedridden status, while rugs at the top indicate participants with bedridden status at three months. (**A**) Calibration of the preoperative model on the development set. (**B**) Calibration of the preoperative model on external validation. (**C**) Calibration of the predischarge model on the development set. (**D**) Calibration of the predischarge model on external validation.

**Figure 3 diagnostics-14-00804-f003:**
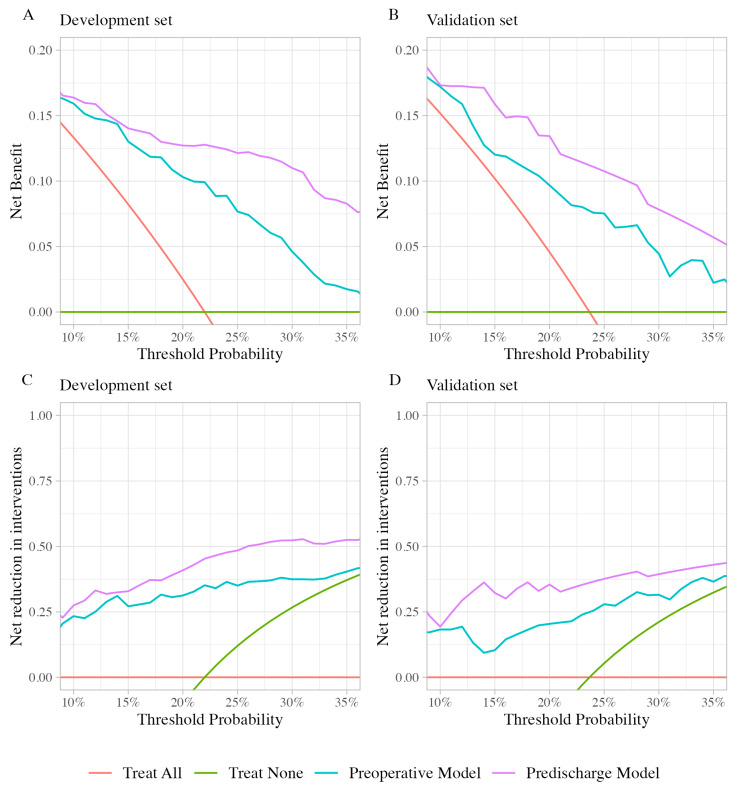
Net benefit curve and net reduction in interventions curve when using the prediction model at the 10–35% cut-off threshold. (**A**) Net benefit on the development set. (**B**) Net benefit on the validation set. (**C**) Net reduction in interventions on the development set. (**D**) Net reduction in interventions on the validation set.

**Figure 4 diagnostics-14-00804-f004:**
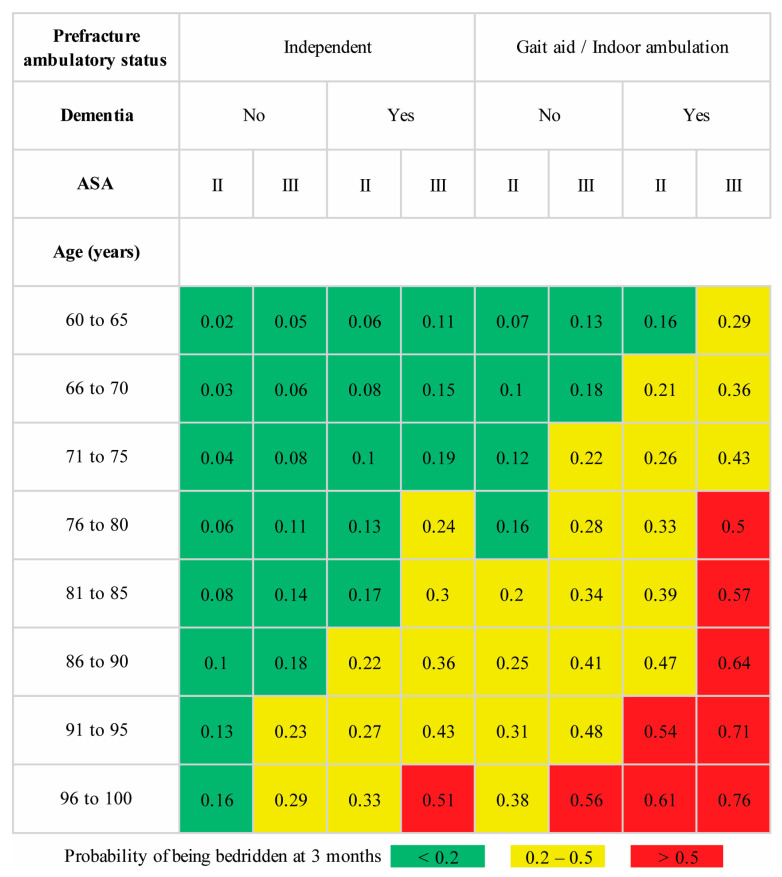
Prediction score chart for the preoperative model for predicting bedridden status at three months after operative treatment of a fragility intertrochanteric fracture with a cephalomedullary nail. The ASA physical status classification is indicated by II (mild systemic disease) and III (severe systemic disease).

**Figure 5 diagnostics-14-00804-f005:**
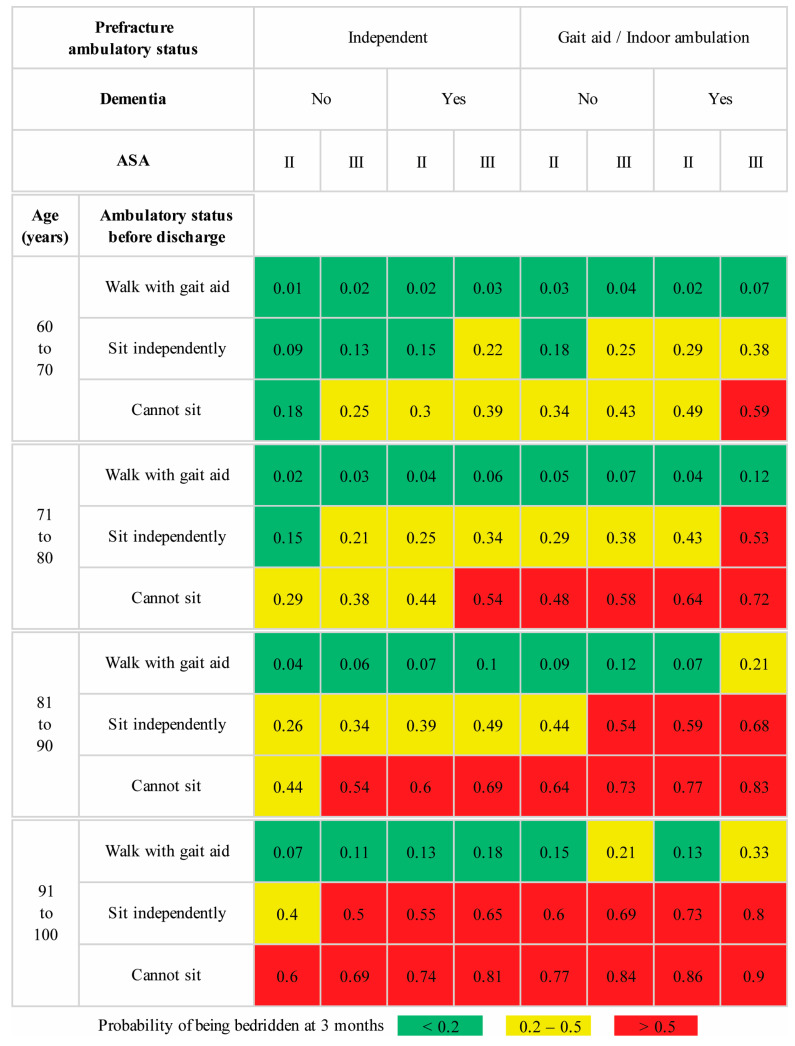
Prediction score chart for the predischarge model for predicting bedridden status at three months after operative treatment of a fragility intertrochanteric fracture with a cephalomedullary nail. The ASA physical status classification is indicated by II (mild systemic disease) and III (severe systemic disease).

**Table 1 diagnostics-14-00804-t001:** Demographic and characteristics of all patients, comparing those in the development and validation sets.

Characteristic	Overall (*n* = 384) ^a^	Development (*n* = 291) ^a^	Validation (*n* = 93) ^a^	*p*-Value ^b^
Age, yrs	83 (78, 87)	83 (77, 86)	85 (79, 89)	0.005
Male	117 (30%)	87 (30%)	30 (32%)	0.7
Body mass index, kg/m^2^	22.1 (19.5, 25.4)	22.0 (19.5, 25.3)	22.9 (19.9, 25.7)	0.5
ASA Classification				0.4
II	141 (37%)	110 (38%)	31 (33%)	
III	243 (63%)	181 (62%)	62 (67%)	
Pre-fracture ambulatory status				<0.001
Totally independent	179 (47%)	151 (52%)	28 (30%)	
Gait aid/Indoor independent	205 (53%)	140 (48%)	65 (70%)	
Comorbidities				
Hypertension	260 (68%)	191 (66%)	69 (74%)	0.12
Diabetes mellitus	102 (27%)	75 (26%)	27 (29%)	0.5
Ischemic heart disease	38 (9.9%)	25 (8.6%)	13 (14%)	0.13
Heart failure	21 (5.5%)	16 (5.5%)	5 (5.4%)	>0.9
Cerebrovascular disease	73 (19%)	52 (18%)	21 (23%)	0.3
Lung disease	40 (10%)	28 (9.6%)	12 (13%)	0.4
Dementia	29 (7.6%)	24 (8.2%)	5 (5.4%)	0.4
Chronic kidney disease	51 (13%)	36 (12%)	15 (16%)	0.4
AO/OTA 2018 classification				0.011
A1	116 (30%)	76 (26%)	40 (43%)	
A2	205 (53%)	168 (58%)	37 (40%)	
A3	38 (9.9%)	29 (10.0%)	9 (9.7%)	
B3	25 (6.5%)	18 (6.2%)	7 (7.5%)	
Time to surgery < 72 h	286 (74%)	205 (70%)	81 (87%)	0.001
Postoperative length of stay, days	5 (4, 8)	5 (4, 8)	5 (4, 8)	0.4
Ambulatory status before discharge				>0.9
Walk with walker	239 (62%)	181 (62%)	58 (62%)	
Sit independently	122 (32%)	92 (32%)	30 (32%)	
Cannot sit independently	23 (6.0%)	18 (6.2%)	5 (5.4%)	
Bedridden at three months	86 (22%)	64 (22%)	22 (24%)	0.7

^a^ Median (IQR); *n* (%), ^b^ Wilcoxon rank sum test; Pearson’s chi-square test. ASA, American Society of Anesthesiologists; AO/OTA the AO Foundation/Orthopaedic Trauma Association.

**Table 2 diagnostics-14-00804-t002:** Specifications of the prediction models.

	Coefficients	95% CI ^a^	*p*-Value	VIF ^a^
Preoperative model				
Intercept	−7.39	−11.4, −3.72	<0.001	
Age, yrs	0.06	0.02, 0.11	0.011	1.02
ASA III	0.72	0.06, 1.42	0.037	1.01
Dementia	0.94	0.03, 1.85	0.042	1.02
Pre-fracture gait aid use/indoor ambulation	1.14	0.50, 1.81	<0.001	1.04
Predischarge model				
Intercept	−8.65	−13.1, −4.62	<0.001	
Age, yrs	0.06	0.02, 0.11	0.009	1.03
ASA III	0.40	−0.36, 1.20	0.3	1.03
Dementia	0.62	−0.39, 1.63	0.2	1.03
Pre-fracture gait aid use/indoor ambulation	0.81	0.10, 1.55	0.028	1.02
Ambulation status before discharge				1.23
Able to walk with a walker (ref)	-	-	-	
Sit independently	2.12	1.39, 2.90	<0.001	
Cannot sit independently	2.95	1.79, 4.21	<0.001	

^a^ CI = confidence interval, VIF = variance inflation factor. ASA, American Society of Anesthesiologists classification.

**Table 3 diagnostics-14-00804-t003:** Discriminative ability of the prediction models.

Performance	AUC	95% CI ^a^
Preoperative model		
Apparent	0.76	0.68, 0.83
Internal validation ^b^	0.74	0.68, 0.80
External validation ^b^	0.72	0.61, 0.83
Predischarge model		
Apparent	0.85	0.79, 0.91
Internal validation ^b^	0.84	0.79, 0.89
External validation ^b^	0.83	0.74, 0.92

^a^ CI = confidence interval, ^b^ validation performance from 1000 bootstrap samples. AUC, area under receiver operating characteristic curve.

## Data Availability

The datasets used in this study are available from the corresponding author upon request. The code that produced the findings of this study is available upon request from the corresponding author.

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
