# Peer review of "Development and Validation of Two-Step Prediction Models for Postoperative Bedridden Status in Geriatric Intertrochanteric Hip Fractures"

_diagnostics, 2024, doi:10.3390/diagnostics14080804_

Round 1

Reviewer 1 Report

Comments and Suggestions for Authors

Development and validation of two-step prediction models for postoperative bedridden status in geriatric intertrochanteric hip fractures. described 2 models to predict the bedridden status 3 months after the hip fracture surgery. The methods were developed based on the patients’ data in the hospital during 2010-2020. Then they validated their models with their recent patients’ data. It is important in the situation when the recovery resources and financial situation are limited and cannot cover all the patients. This method is supposed to be a 2-step process as mentioned in the title and abstract. It is very confusing that the two-step method is replaced by 2 individual models in the result and method part. I didn’t find any relationship between the 2 individual models during the prediction. Maybe the prediction curve analysis is the second step? Please confirm in the paper.

The net benefit curve part is unclear. Please introduce the net benefit and then make clear how to make the benefit curve. We do not have the equation or any information to understand the 2 models that authors described, therefore we cannot understand how to make the curve. And it is important for me to evaluate the curve. The paper mentioned “treat all policy” and “treat non policy” but did not systematically analyze or describe in the paper. Please explain them in the result part, including the meaning of their curves. They are important controls and critical for understanding the meaning of the 2 models. The figure legend for the net benefit curve doesn’t give enough information to support the understanding. The legend should be able to explain all the details in the figure. The annotation on the figure is also not detailed enough. Therefore, all these make the reading and understanding very hard.

Patients w/ or w/o recovery training or intervention should be documented in those patients in both development process and validation process. It is interesting to see if the treatment or intervention can really help patients to avoid the bedridden status. Should the medical intervention after discharge should be considered in these 2 models?

Reviewer 2 Report

Comments and Suggestions for Authors

This innovative and clinically instructive study creates a two-step prediction model using simple, easily retrievable predictors reported in the literature to assist clinicians in deciding on care during hospital stay and after discharge. Classifying high-risk patients can help optimize resource allocation, reduce the cost of implementing rehabilitation programs, and alleviate the workload on healthcare workers by minimizing unnecessary interventions for low-risk patients. However, there are still some issues that require further clarification

1. The selection of predictors in this study did not include choice of anesthesia, time to surgery after a fall, or nutritional status. Are these factors also critical?

2. Are there ethical issues in the validation phase of this study? Is it based on active interventions for high-risk patients?

3. It is recommended to increase the number of validation cohorts.

4. Is using prosthetic arthroplasty in elderly patients more likely to promote rapid recovery?

5. This is a retrospective study, and there may be some bias in case selection and data processing.

Reviewer 3 Report

Comments and Suggestions for Authors

This paper discusses an practical issue. The topic that has been conferred about in the paper is part of a comprehensive discussion on the problem of postoperative bedridden status in geriatric intertrochanteric hip fractures.

While I think this is an interesting topic, the manuscript could be improved.

Introduction

The study lacks of clearly defined goal and research questions.

Material and Methods

The authors have to explain:

In what country was this study conducted? Where are these people from? From Europe, America, Asia…

The inclusion and exclusion criteria require clarification.

Statistical Analysis

Was the consistency of the results with the normal distribution tested?

If so, the authors have to explain: All variables were consistent with the normal distribution?

If the normality criteria are not met the authors should use the median, otherwise the can opt for the mean.

Results

The analysis of the results is unclear. Please describe the results in more detail.

Discussion

In this part, the data or main findings in you study should be compared with the another  researchs.

Conclusions

Please specify the conclusions in points as answers to the research questions.

General comments to the Authors

That said, my comments are offered with the intent of helping the authors improve this manuscript. When the authors address these issues I will be able to comment definitively and make the final decision.

Round 2

Reviewer 1 Report

Comments and Suggestions for Authors

The manuscript was improved dramatically and addressed all my concerns. Please show the meaning of the net benefit and net reduction. Please also show how to calculate the net benefit and reduction in the result or method part. 

Usually, people do not claim that they developed a successful model. The other people will judge whether the model is successful. 

Comments on the Quality of English Language

The English is fine.

Reviewer 3 Report

Comments and Suggestions for Authors

Not all of the reviewer's comments have been taken into account, so I am asking for corrections on the following issues:

Introduction

The study is still out research questions (or hyphotesis).

Material and Methods

The inclusion and exclusion criteria require clarification.

Discussion

In this part, the data or main findings in you study should be compared with the another  researchs, but results should not be included in the Discussion. Please rewrise Disscussion.

Conclusions

Please specify the conclusions in points as answers to the research questions.

General comments to the Authors

Please describe in bullet points all corrections and additions made.

When the authors address these issues I will be able to comment definitively and make the final decision.

Round 3

Reviewer 3 Report

Comments and Suggestions for Authors

Accept in present form